# Assessment of Predictors for SARS-CoV-2 Antibodies Decline Rate in Health Care Workers after BNT162b2 Vaccination—Results from a Serological Survey

**DOI:** 10.3390/vaccines10091443

**Published:** 2022-09-01

**Authors:** Nadav Zacks, Amir Bar-Shai, Hezi Levi, Anna Breslavsky, Shlomo Maayan, Tsyba Evgenia, Shlomo Feitelovich, Ori Wand, Moshe Schaffer, Yaniv Sherer, Gili Givaty, Anat Tzurel Ferber, Tal Michael, Natalya Bilenko

**Affiliations:** 1Faculty of Health Sciences, Ben-Gurion University of the Negev, Beer-Sheva 84101, Israel; 2Pulmonary Department, Barzilai University Medical Centre, Ashkelon 7830604, Israel; 3Division of Infectious Diseases, Barzilai University Medical Centre, Ashkelon 7830604, Israel; 4Laboratory Division, Barzilai University Medical Centre, Ashkelon 7830604, Israel; 5Department of Oncology, Barzilai University Medical Centre, Ashkelon 7830604, Israel; 6Medical Administration Department, Barzilai University Medical Centre, Ashkelon 7830604, Israel; 7Regional Medical Office of Ministry of Health, Ashkelon District, Ashkelon 7830604, Israel

**Keywords:** SARS-CoV-2 antibodies, Barzilai Medical Center, Health-Care-Works, BNT162b2 vaccination, demographic factors, vaccination regimens, waning of antibodies, booster vaccinations, patiant contect

## Abstract

Background: SARS-CoV-2 is a novel human pathogen causing Coronavirus Disease 2019 that has caused widespread global mortality and morbidity. Since health workers in Israel were among the first to be vaccinated, we had a unique opportunity to investigate the post-vaccination level of IgG anti-S levels antibodies (Abs) and their dynamics by demographic and professional factors. Methods: Prospective Serological Survey during December 2020–August 2021 at Barzilai Medical Center among 458 health care workers (HCW) followed for 6 months after the second BNT162b2 vaccine dose. Results: Antibody levels before the second dose, and 30, 90 and 180 days after were 57.1 ± 29.2, 223 ± 70.2, 172.8 ± 73.3 and 166.4 ± 100.7 AU/mL, respectively. From GEE analysis, females had higher Abs levels (β = 26.37 AU/mL, *p* = 0.002). Age was negatively associated with Abs, with a 1.17 AU/mL decrease for each additional year (*p* < 0.001). Direct contact with patients was associated with lower Abs by 25.02 AU/mL (*p* = 0.009) compared to working with no such contact. The average decline rate overall for the study period was 3.0 ± 2.9 AU/mL per week without differences by demographic parameters and was faster during the first 3 months after vaccination than in the subsequent 3 months. Conclusions: All demographic groups experienced a decline in Abs over time, faster during the first 3 months. Findings of overall Abs lower in males, workers with direct contact with patients, and older workers, should be considered for policy-making about choosing priority populations for additional vaccine doses in hospital settings.

## 1. Introduction

SARS-CoV-2 is a novel human pathogen causing Coronavirus Disease 2019 (COVID-19). COVID-19 is a devastating disease that has caused widespread global mortality and morbidity [1,2]. COVID-19 was declared a pandemic by the World Health Organization (WHO) on 11 March 2020 [3]. Severe disease may result in respiratory failure and multiorgan dysfunction, leading to long-term sequela in recovered patients [4]. Until December 2021, there were over 250 million confirmed cases and over 5 million deaths worldwide. Israel has over 1.3 million confirmed cases and over 8000 deaths [5,6].

Antibodies, including IgM, IgA, and IgG, can be detected in the blood 5 to 15 days following symptom onset or a positive Reverse Transcriptase Polymerase Chain Reaction (RT-PCR) test, with IgM typically appearing first [7]. Specific IgG antibodies to SARS-CoV-2 antigens can be detected long after the resolution of the infection [8].

The pandemic has prompted the rapid development of vaccines. On December 2020, the U.S. Food and Drug Administration (FDA) granted an Emergency Use Authorization for the Pfizer-BioNTech BNT162b2 mRNA vaccine [9]. On 23 August 2021, the FDA approved the BNT162b2 vaccine for persons aged ≥16 years [10]. The BNT162b2 vaccine promotes the generation of IgG antibodies against viral spike glycoproteins (S1 and S2). The initially recommended vaccination regimen included two intramuscular injections, given 21 days apart. The FDA has approved a third “booster” shot that may be administrated 5 months later due to concerns of waning immunity over time [11,12,13].

The phase-3 study of BNT162b2 validated the efficacy and safety of the vaccine: two doses of BNT162b2 provided 95% protection against COVID-19 in persons 16 years and older, and the safety was similar to other viral vaccines over an average of two months [14]. The duration of protection, including the rate at which the anti-SARS-CoV-2 antibodies decrease, is not fully known. Several studies have reported the humoral response in vaccinated subjects over various periods [15], some of which focused on groups with specific medical conditions.

Several other studies demonstrated that antibody levels were dynamic after the second dose [16] and differed by gender and age [17,18,19,20]. 95.5% of participants developed anti-spike antibodies after 14 days, and antibody titers increased 24.9-fold after the second dose. As part of our study, we examined additional risk factors for postvaccination immunity waning in a multisectoral hospital setting.

In this study, we have investigated the dynamics of post-vaccination antibody levels over time in health care workers (HCW) of Barzilai Medical Center (BMC) and examined the effects of sociodemographic and professional factors. On 19 December 2020, Israel began a national vaccination campaign that prioritized individuals at high risk due to chronic medical conditions, HCWs, and seniors.

Understanding factors associated with peak antibody levels and the rate of decline may prove significant in planning future vaccination regimens. They may enable policymakers to create a better protection plan for specific population groups. It is still debatable which dosing regime is optimal and which populations will be most affected by additional doses. Understanding variables that affect the immune response to vaccination and the rate of decline may allow custom vaccination plans to be created for individuals requiring prioritization.

We aimed to assess the humoral response to the BNT162b2 vaccine in HCW over time and to identify sociodemographic and professional factors influencing the waning of antibodies after the second dose.

## 2. Methods

A prospective serological survey was conducted from December 2020–August 2021 among HCWs of BMC, a tertiary hospital in Ashkelon, Southern Israel, serving approximately 600,000 people.

### 2.1. Study Population

Barzilai employs 2700 HCWs, of which 90% (2430 workers) have received at least two doses of the BNT162b2 vaccine at the time of data collection. Twenty percent (20%) of HCWs do not have direct professional contact with patients. Inclusion criteria were an age of 18 or older, signing of an informed consent form, receiving at least two doses of the BNT162b2 vaccine, and having serological tests performed 0-5 days prior and 30 days after the second dose of the vaccine. HCWs with a history of infection with SARS-CoV-2, either before or during the serological survey, were excluded.

Of 2430 HCWs, 600 were eligible for the study according to the study protocol and were offered to participate in the serological survey. Out of the 600 eligible HCW, 458 (76%) have agreed to participate in this study. Four hundred fifty-two were tested 0-5 days before the second dose of the vaccination (Baseline Test), 355 were tested 28 ± 6 days after the second vaccine dose (Test #1), 263 were tested 98 ± 7 days after the second dose (Test #2), and 289 were tested 194 ± 7 days after the second dose (Test #3) (Figure 1). At Test #1, 343 (96.6%) participants were asked to fill out a demographic and vaccine-related side effects questionnaire. At the third serological test (Test #3), participants filled out a questionnaire to determine their physical activity habits, including chronic medication usage (both prescribed medication and off-the-counter). This information was available for 189 participants. At each time point, we evaluated the participants for SARS-CoV-2 infection and excluded exposed individuals (36 excluded overall; Baseline Test: 18 participants; Test #1: 4 participants; Test #2: 8 participants; Test #3: 6 participants). After that, 181 participants completed all tests and tested negative for SARS-CoV-2 during the survey. Statistical analysis was performed on the participants who were found negative for COVID-19 exposure, regardless of how many tests they had participated in (N = 422), except for the IgG Anti-S decline rate calculation, which was calculated on the participants who completed all the tests and were found negative for COVID-19 exposure to the (N = 181).

### 2.2. Study Procedures

Antibody levels against the viral spike antigen (IgG anti-S) were assessed at 4 predetermined time points: 0-5 days before the second dose of the vaccination (Baseline Test) and 30, 60, and 180 days after the second dose (Tests #1, #2, and #3, respectively). The participants were tested for COVID-19 blood S1/S2 IgG type antibodies (Abs), using Liaison chemiluminescent immunoassay kit (DiaSorin, Saluggia, Italy; REF 311450) to assess immunological status after vaccination.

Infection with SARS-CoV-2 was diagnosed either by a positive RT-PCR in a nasal swab or, retrospectively, by a positive anti-nucleocapsid antibody test (anti-N). All participants underwent two tests for anti-N antibodies on each occasion where IgG anti-S were measured, using both the Elecsys Anti-SARS-CoV-2 immunoassay (Roche Diagnostics, Basel, Switzerland) on a Cobas analyzer and the Abbott SARS-CoV-2 IgG nucleocapsid protein assay (Abbott, Abbott Park, IL) on an Architect analyzer.

### 2.3. Statistical Analysis

Dependent variables used in this study were (1) geometric means of IgG anti-S antibodies levels at each stage and (2) the rate of Abs levels decline (RD) as the difference between levels of antibodies between different tests per week. Independent variables used in this analysis were: age as a continuous variable and age as a dichotomous variable (age ≤ 50 years, age > 50 years), sex (male/female), place of birth (Israel, Asia/Africa, Europe), profession (direct contact with patient [such as physicians, nurses, radiologists, physiotherapists, etc.] and no direct contact with patient [administrators, laboratory workers, housekeepers, IT workers, etc.]), obesity (BMI > 25), self-reported smoking status (ever/never), self-reported overall physical activity (at work and recreational—yes/no), night shifts (yes/no), self-reported regular use of medications for chronic conditions (yes/no). We used ANOVA for comparing Abs between categorical variables and Chi-square or Fisher’s Exact test when appropriate for comparing proportions. We tested associations between Abs and independent variables to detect possible confounders. Finally, multivariate linear regression analysis was built to detect the unique and independent effect of factors contributing to changes in Abs at each Test. Following this stage, generalized estimating equations (GEE) models were conducted, accounting for each participant’s repeated measures. The goodness of fit was examined using Bayesian information criteria (BIC), and the most fitted models were chosen. Betas and 95% confidence interval (CI) were calculated. Statistical significance was set at *p* ≤ 0.05. Data analysis and statistical procedures were conducted using SPSS 25.0 ^®^ (SPSS, Chicago, IL, USA), MATLAB 2021 ^®^ software, and R statistics (version 4.1.1; A Language and Environment for Statistical Computing, Vienna, Austria)), including the R packages: ‘data.table’, ‘ggplot2’, ‘dplyr’, ‘lubridate’ and ‘gee’.

### 2.4. Ethical Considerations

The study was approved by the Ethics Committee and Institutional Review Board of Barzilai Medical Centers (No. 0009-21-BRZ). The study was performed in accordance with the Declaration of Helsinki and Good Clinical Practice guidelines. All participants signed an informed consent form prior to study enrollment. Results are reported according to STROBE statement guidelines.

## 3. Results

The study population included 458 HCWs from BMC belonging to different medical, paramedical, and management sectors. Our study population generally included relatively young (48.7 ± 10.7 years old), mostly female HCWs (Table 1). Fifty-nine percent of participants had self-reported BMI above 25. Only one-fifth were Israeli-born. Over half of studied HCWs have direct contact with patients during work.

We examined levels of IgG anti-S Abs after the first and second vaccine doses and the rate of decline per week over time. At the baseline test, performed 0–5 days prior to the second vaccine dose, the average antibody levels were 57.1 ± 29.2 AU/mL (median 55.8), with no statistically significant difference between males and females.

Four weeks (28 ± 6 days) after the second dose (Test #1), the antibody levels increased to an average of 223 ± 70.2 AU/mL (Median 214). Ninety days after the second dose of the vaccination (Test #2), antibody levels declined to a mean of 172.8 ± 73.3 AU/mL (median 160). At the last measurement performed six months after the second vaccine dose (Test #3), Abs continued to decrease to an average of 166.4 ± 100.7 AU/mL (Median 137.5).

In the univariate analysis, we found statistically significant associations between IgG anti-S levels and sex, age, origin, type of contact with a patient, and physical activity, but not with obesity status, working night shifts and smoking (Appendix A, Table A1).

Beginning from Test #1, female participants had higher levels of IgG anti-S, although this difference was statistically significant only for Test #1 (*p* = 0.036, 0.416 and 0.163 for Tests #1, #2 and #3 respectively) (Figure 2A). At all tests, IgG anti-S Abs levels were higher for younger individuals. This difference was pronounced and statistically significant when comparing participants 50 years old or younger with older participants (*p* = 0.041, 0.088, and 0.557 for Tests #1, #2 and #3, respectively). However, the negative association of antibody levels with age was less pronounced between tests (Figure 2B). Health care workers who have direct contact with patients had consistently lower antibody levels than those without such contact starting from Test #1 but less significantly at Test #3 (*p*= 0.018, 0.015, and 0.082 for Tests #1, #2, and #3, respectively) (Figure 2C).

Further, we examined the rate of Abs decline calculated as the IgG anti-S in AU/mL difference between tests divided by the number of weeks between them. The average weekly decline rates were 5.3 ± 4.31 AU/mL/week (median 5.1) between Test #1 and Test #2 and 0.84 ± 3.1 AU/mL/week (median 1.5) between Test #2 and Test #3. The average decline rate during the overall study period was 3.0 ± 2.9 AU/mL/week (Median 3.1). We did not find a significant association between sociodemographic parameters and the rate of decline.

To identify background variables that could have influenced the humoral response to vaccination, we developed a GEE model that incorporated variables significantly associated with IgG anti-S levels as univariates. We found an independent negative association between age in years and IgG anti-S levels with a β of -1.17 for each additional year (*p* < 0.001). Since the association was similar for age groups 50 years old or younger and older than 50, we also assessed age as a categorical variable in our model. We found that antibody levels are almost 20 AU/mL lower in individuals over 50 years old compared to younger ones (*p* = 0.013). Female gender and working without direct patient contact were found to be associated with higher antibody levels (*p* = 0.002, 0.009, respectively) (Table 2).

## 4. Discussion

In this prospective study, we aimed to study how sociodemographic and professional characteristics influence the titers and decline rates of IgG anti-S antibodies in HCWs vaccinated with two doses of BNT162b2. Like others [21,22], we found a significant decrease in antibody levels over time in all demographic groups. We found statistically significant associations between IgG anti-S levels and sex, age, and type of contact with a patient, but not with obesity status, physical activity, origin, working night shifts and smoking. These associations were more significant at the beginning of the survey.

In addition to assessing the humoral response to the BNT162b2 vaccine, our study also aimed to investigate the influence of sociodemographic factors on the rate of decline in antibody levels. To our knowledge, this is the first time the decline rate has been measured and described. We found that absolute Abs levels were negatively associated with age and male sex of HCW while working in direct contact with patients related to lower levels. However, we have found that sociodemographic variables do not affect the average rate of decline over 6 months. Nevertheless, we found that the decline rate in the first 3 months after the second dose was 5–7 times faster than during the following 3 months. Compared to non-mRNA-based vaccines, antibodies’ decline rates are considerably slower, with no evidence of significant change in the rate during the first years after the vaccine [23], suggesting that the humoral response to the BNT162b2 vaccine may be less durable. One advantage of mRNA vaccines is their short life with an early shut-down of antigen expression [24]. This aspect of the promising vaccine may be the Achilles’ heel of the technology.

We developed a model to assess the individual effects of different variables on IgG anti-S levels over time. Surprisingly, we identified work environments without exposure to patients as associated with a lesser decline in antibody levels over time. This observation is possibly explained by the fact that employees exposed to patients are also more exposed to various infectious diseases; thus, their immune system is engaged in multiple processes, making it less available for the short-lived BNT162b2 vaccine.

Uwamino et al. [16] reported similar findings among Japanese participants working in a medical school and its affiliated hospital in Tokyo, for whom serum samples were collected before the first dose and three weeks after the second dose of the BNT162b2 vaccine. They measured antibody titers against the receptor-binding domain of the spike protein of SARS-CoV-2. They found that young age (<45 years), female sex, and adverse reactions after the second dose were independently related to higher antibody titers after the second dose. Similar findings were reported in other studies [18,19,20].

Limitations of the study include a modest sample size and high rates of participants lost to follow-up or excluded due to SARS-CoV-2 infection over time, which could have biased the results. Humoral responses are commonly used as surrogate markers for immunity after vaccination. Measuring antibody levels is easier to perform and standardize, less expansive, and commercially available compared to assessing antibodies’ cellular immune responses or binding capacities. Yet, the clinical implications of antibody levels are indeterminate at best. Several publications reported that post-vaccination antibody titers are predictive of immune protection from COVID-19 [25,26,27], and real-life studies reported that breakthrough infections correlated with low antibody titers before infection in HCW [28] and hemodialysis patients [29], yet a protective threshold antibody level for immunity in currently undefined.

While the initial efficacy of mRNA-based vaccines, including BNT162b2, against contracting SARS-CoV-2, and especially against severe COVID-19, is remarkably high, it is apparent that this efficacy declines over time. Our results strengthen the indispensability of additional vaccine doses. Additional vaccine doses, “boosters,” confer improved and longer-term protection from disease. However, the optimal dosing regime is still debated, and so are the target populations which will benefit the most from additional doses. A better understanding of variables that affect the immune response to vaccination may allow personally tailored vaccination schedules for individuals requiring prioritization. Moreover, the fast rate of decline observed in our cohort raises the need to develop a more efficient and long-lasting COVID-19 vaccine.

## 5. Conclusions

Female sex, younger age, and no direct contact with patients is associated with higher IgG anti-S levels in vaccinated HCW. All demographic groups experienced a steep decline in antibody levels over time. The decline rate of antibody levels is significantly faster during the first months after the vaccination and slows after that. We believe this study will be used as grounds for further exploration of the differences in IgG anti-S levels in HCWs with direct contact with patients from HCWs without direct contact. Considering the variables that affect absolute antibody levels and the rate of decline may be the key to deciding when to prioritize the administration of additional vaccine doses in HCW. This group differs from the general population in terms of age, health status, activity, sex distribution, and close, continuous contact with in-hospital patients.

## Figures and Tables

**Figure 1 vaccines-10-01443-f001:**
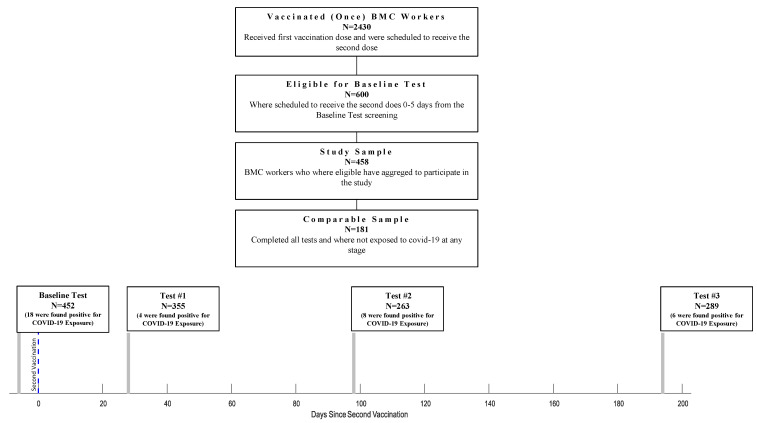
Flow chart of HCW who Participated at Each Stage of The Survey.

**Figure 2 vaccines-10-01443-f002:**
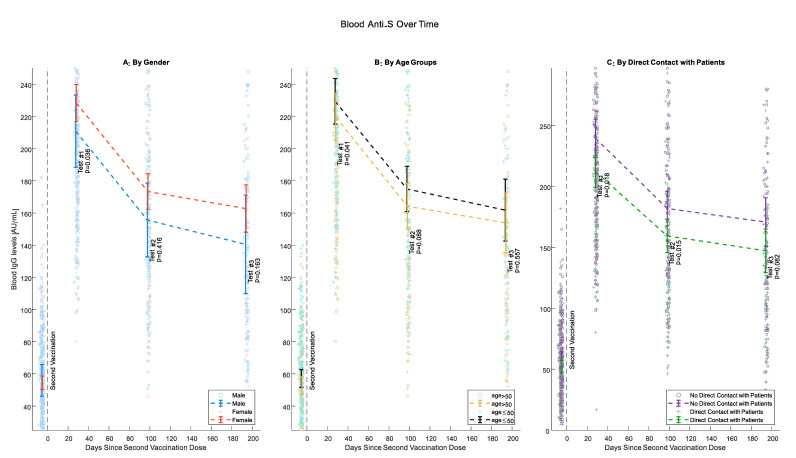
IgG anti-S Levels Over Time. (**A**) Gender; (**B**) age (50 or younger, Older than 50); (**C**) direct patient contact. Circles and crosses represent individual IgG Anti-S values; means and 95% confidence intervals are only calculated for the comparable sample.

**Table 1 vaccines-10-01443-t001:** Demographic characteristics of the study population.

Variable	n (%) or Mean ± SD (Median)
**Sex**	N = 458
Male	128 (28%)
Female	330 (72%)
**Age** [years]	48.7 ± 10.7 (50%)
**Place of Birth**	N = 434
Israel	93 (21.4%)
America/Europe	128 (29.5%)
Asia/Africa	213 (49.1%)
**Chronic use of medications**	N = 186
Yes	74 (40%)
No	112 (60%)
**Physical activity**	N = 189
Yes	110 (58.2%)
No	79 (41.8%)
**Direct professional contact with a patient**	N = 434
Yes	235 (54.1%)
No	199 (45.9%)
**BMI**	26.0 ± 23.4 (29.4)
**Smoker (active or past)**	N = 435
Yes	292 (67.3%)
No	142 (32.7%)
**Night shifts**	N = 187
Yes	134 (72%)
No	53 (28%)

Some data was not available for all participants.

**Table 2 vaccines-10-01443-t002:** Independent effects of sociodemographic and personal background variables on levels of antibodies from the GEE model.

Characteristic	Estimate	Standard Error	*p*-Value
Test Number	−32.28	2.39	<0.001
**Age**			0.013
≤50 years	Ref.		
>50 years	−19.92	7.10	
**Sex**			0.002
Male	Ref.		
Female	26.37	8.13	
**Type of contact with patients**			0.009
Direct contact	Ref.		
Nondirect contact	25.02	7.09	

## Data Availability

The data presented in this study are available on request from the corresponding author. The data are not publicly available in accordance with this study’s ethical protocol.

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
