# Peer review of "Assessment of Predictors for SARS-CoV-2 Antibodies Decline Rate in Health Care Workers after BNT162b2 Vaccination—Results from a Serological Survey"

_vaccines, 2022, doi:10.3390/vaccines10091443_

Round 1

Reviewer 1 Report

  • The aim of this study is to assess the humoral response to the BNT162b2 vaccine over time. This is a relevant question in the context of this pandemic time but It is known that the antibody wane over time so this is the reason to administer booster dose. This study provides more evidence in this regard
  • General concept comments

    The article is well written and it's clear and understandable so i don't have more comments to do. I am surprised that those who have contact with patients have lower levels of antibodies.Have you analyzed whether there was some difference between the group that had contact with patients and those that not?
  • Specific comments: reference 24 is incomplete, lines 167-169

Author Response

Please see the attachment, thank you

Reviewer 2 Report

The manuscript titled Assessment of Predictors for SARS-CoV-2 Antibodies Decline Rate in Health Care Workers after BNT162b2 Vaccination - Results from a Serological Survey by Zacks et al. investigated the post-vaccination level  of IgG anti-S levels antibodies (Abs) and their dynamics by demographic and professional factors in a unique population from the healthcare workers of Barzilai Medical Center. This study describes some interesting observations, old people and males have relatively lower Abs within 6 months after the second BNT162b2 vaccine dose, which is consistent with other studies such as Y. Uwamino, T. Kurafuji, Y. Sato et al. 2022, https://doi.org/10.1016/j.vaccine.2022.01.002. In addition,  there is another interesting observation- healthcare workers with direct contact with patients have significantly lower Abs. However, it’s not sure if this is contributed by age and sex. It would be more interesting to have a look at the difference between the two groups by balancing age and sex. This manuscript is well organized. Some  minor suggestions are listed below:

In abstract: azdditional should be checked?  “All demographic groups experienced a decline in Abs over time, faster during the first 3 months. Findings of overall Abs lower in males, workers with direct contact with patients and older workers”, need additional comparison for this statement.

Introduction. The information described is too general and it can be more specific for this study and move the “Study Goals” in this section. Check typos.

Methods. How to control subjects with a history of immunodeficiency and other health conditions? What is the assay catalog number? What means “(Error! Reference source not found.)”? Is missing data or something else? The summary information for TEST#1,#2,#3 etc needs to be described more concisely or some information can be included into Supplemental table 1.

Results. To increase the font size for Figure 1. A, B, C separated from the plot title or removed; in Table 1 for Smoker (active or past) *, the sample size is 414 or 189?  Demographics replacing Background in the title would be more properly;  In Table 2, < 50 years vs 50+, format as < 50 years vs >50 or 50- vs 50+; “Test Number” means intercept?

Discussion. How comparable of the observations in this study compared with another similar study., e.g Uwamino et al., 2022. Check typos everywhere.

“Since the association was similar for ages groups 50 years old or younger and older than 50 years old, we also assessed age as a categorical variable in our model and found that antibody levels are almost 20 AU/ml lower in individuals over 50-years-old compared to younger ones (p=0.013).” need to be checked. The Abs number scale is different from that in Uwamino et al 2022, check AU/ml vs IU/ml.

Conclusions. Female sex, younger age, and no direct contact with patients are associated with higherIgG anti-S levels in vaccinated HCW. Check error. More exploration for no direct contact vs direct contact is needed.

Author Response

Please see the attachment, thank you
